# Is skin disinfection before subcutaneous injection necessary? The reasoning of Certified Nurses in Infection Control in Japan

Yuko Yoshida [1]*, Risa Takashima[2], Rika Yano[1]

1 Department of Fundamental Nursing, Hokkaido University Faculty of Health Sciences, Sapporo, Hokkaido, Japan, 2 Department of Functioning and Disability, Hokkaido University Faculty of Health Sciences, Sapporo, Hokkaido, Japan

These authors contributed equally to this work.

* yuko790402@hs.hokudai.ac.jp

**Data Availability Statement:** The data for this study consists of transcripts of interviews with 10 participants and contain identifying items, and are

## Abstract

Nurses continue to disinfect the skin before administering subcutaneous injections as a standard process in clinical settings; despite evidence that disinfection is not necessary. To implement evidence-based practice, it is critical to explore why this gap between "evidence" and "practice" exists. This study aimed to describe the reasons offered by Certified Nurses in Infection Control (CNIC) in Japan for performing skin disinfection before subcutaneous injection. Adopting an inductive qualitative design, interviews were conducted with 10 CNIC in 2013. According to the participants, skin disinfection before subcutaneous injection: (a) was common practice; (b) may have been beneficial if it was omitted; (c) adhered to hospital norms; (d) prevented persistent suspicion of infection; (e) had no detrimental effect; (f) was an ingrained custom; and (g) involved a tacit approval for not disinfecting in home care settings. The themes (c) and (g) were cited as the main reasons affecting decision-making. The CNIC administered injections following skin disinfection in hospitals in accordance with hospital norms. On the contrary, outside the hospital, they administered subcutaneous injections without skin disinfection. All themes except (b) and (g) reflect the barriers and resistance to omitting skin disinfection, while (g) shows that it is already partly implemented in home care settings. It is necessary to create a guideline for skin disinfection before subcutaneous injection that considers the quality of life of patients at home, their physical conditions, and the surrounding environment at the time of injection, in addition to the guidelines applicable in hospitals.

## Introduction

Disinfecting the skin before administering a subcutaneous injection is a standard procedure in clinical settings for nurses. The rationale behind this practice is that the needle breaks the skin barrier and increases the risk of introducing an infection [1]. However, the Forum for Injection Technique UK [2] and Tandon et al. [3] report that disinfection before

therefore sensitive to privacy issues. As participants only allowed the interviews under promise of anonymity, we are expressly forbidden by the participants to make the full content of the interviews public. Anonymized excerpts from the full transcripts can be made available to qualified researchers by request to the ethical committee of the Faculty of Health Sciences, Hokkaido University, who can be contacted at shome@hs.hokudai.ac.jp.

**Funding:** This study was supported by the Japan Society for the Promotion of Science, Tokyo, Japan (KAKENHI Grant Number JP26861850).

**Competing interests:** The authors have declared that no competing interests exist.

subcutaneous injection of insulin is not necessary. The World Health Organization (WHO) Best Practices for Injections and Related Procedures Toolkit [4] recommends washing the skin with soap and running water before administering a subcutaneous injection. Although skin that is visibly soiled or dirty must be washed, swabbing a patient's clean skin before giving an injection is unnecessary [5]. Furthermore, disinfection is usually burdensome and not required when injections are given in non-institutional settings such as homes, workplaces, or restaurants [6].

Dunleavy et al. and Hope, Hickman, Parry, and Ncube [7, 8] report that not using an alcohol swab is a risk factor for skin and soft tissue infections. In contrast, in some patients, unnecessary disinfection adds to the time needed for self-care. Patients with diabetes receive subcutaneous injections daily; they are hyperglycemic and have reduced function of various immunocompetent cells [9] and, therefore, need to take daily measures to prevent infection. Pre-injection skin disinfection is thus very important in patients with diabetes who self-inject insulin. However, studies dealing with patients with diabetes who self-inject insulin suggest no increased risk of infection when doses are given without skin preparation. In a pre-test/post-test design, Koivisto and Felig [10] study 13 patients who received over 1,700 insulin self-injections with and without skin preparation. No cases of local or systemic infections were found during the three-to-five-month study period. Similarly, McCarthy, Covarrubias, and Sink [11] study 50 patients who received 1,800 self-injections of insulin in a crossover trial of skin preparation with alcohol or tap water, or with no skin preparation, none of whom experienced injection site complications. Thus, skin disinfection before administration of subcutaneous injections is an unnecessary process, and it could burden patients who may not require disinfection. Fleming, Jacober, Vandenberg, Fitzgerald, and Grunberger [12] reported benefits, such as ease of procedure, when patients with diabetes omitted skin disinfection before administering subcutaneous injections, compared to those who disinfected their skin.

A study conducted in Greece by Theofanidis [13] indicates that nurses disinfect the skin before insulin injections as a longstanding medical ritual, although there is insufficient evidence on the need for disinfection. This is true in other parts of the world as well, including Japan [14]. Nurses who do not have extensive knowledge of infection control may assume that skin disinfection before administration of subcutaneous injections prevents infection. According to recent books published in Japan on nursing techniques and skills, disinfection is necessary, while only a few books have introduced studies verifying that it is unnecessary [15]. In contrast, Certified Nurses in Infection Control (CNIC) receive certification from the Japan Nursing Association for specializing in infection control and having advanced nursing skills. CNIC have more experience, skill, and knowledge-based perceptions than other nurses regarding skin disinfection before subcutaneous injection. In this study, the word "experience" is defined as "practical knowledge, skill, or practice derived from direct observation of or participation in events or in a particular activity" [16], while the word "perception" is defined as "The way in which something is regarded, understood, or interpreted" [17]. Describing CNIC's experience and perception of skin disinfection before subcutaneous injection was considered suitable for this objective.

Although omitting skin disinfection before administering subcutaneous injections is safe, reduces the burden on patients, and likely reduces costs, it has still not become standard clinical practice. Exploring the reasons for this lack of adoption in clinical practice in Japan will help us address obstacles to the introduction of new evidence. Thus, we pose the following research question: why do Japanese nurses disinfect skin before administering subcutaneous injections? Based on this research question, the purpose of this study was to describe, using a qualitative design, CNIC's reasoning for disinfecting or not disinfecting the skin before administering a subcutaneous injection.

## Methods

### Study design

An inductive qualitative design was used. The participants' perceptions and experiences of the phenomena under examination are described under a qualitative descriptive approach [18], which produces a straightforward explanation of participants' experiences in their own words [19]. The aim of a qualitative description is not thick description (ethnography), theory development (grounded theory), or interpretative meaning of an experience (phenomenology), but a rich, straight description of an experience or an event [18].

### Participants and data collection

To meet conditions similar to the skills required for a subcutaneous injection, the participant selection criteria were as follows: (1) worked as a staff nurse with more than 10 years of experience and (2) acquired CNIC qualification. We adopted convenience sampling. The first author contacted the director of nursing at the hospitals in which the CNIC worked. The researchers asked a total of 10 nursing department directors of hospitals with over 300 hospital beds to be introduced to the CNIC by telephone and letter. The director of one hospital's nursing department declined to cooperate because the significance of the study was not understood. Nine directors of hospital nursing departments agreed to participate, and introduced their hospital's CNIC to the researchers one by one. One nursing manager introduced two CNIC from her hospital there are usually only one or two such nurses in a hospital. The researchers visited the 10 CNIC, explained the purpose of the study to them, and sought their voluntary participation. To ensure that the study ideas did not influence the participants, the researchers did not disclose their own views on skin disinfection before subcutaneous injection.

The 10 CNIC agreed to participate in the study. Each of them had 15–26 years of nursing experience and 3–7 years of experience as CNIC. They worked primarily in non-profit hospitals, and seven of them worked in hospitals with bed numbers ranging from 300 to 500. The remaining three nurses worked in 500- to 900-bed hospitals. One of the ten nurses was a ward head nurse, and one was a staff nurse. The other eight were not assigned to a ward, but were assigned to a department for which they worked across the hospital departments as infection control specialists. The CNIC had all worked in several wards as staff nurses for more than 10 years. Therefore, although there were participants who did not intervene frequently in their colleagues' work, all participants were in a position to instruct nurses on infection control measures, including skin disinfection before injection.

Data were collected from August to November, 2013 using individual semi-structured interviews. Data collection and analysis were performed simultaneously. After interviewing five participants individually, analysis and categorizations were made. After each interview, an analysis confirmed the emergence of new categories. Up to the eighth participant, a new category was identified each time; hence, more interviews were conducted with additional participants. Two more participants were interviewed, but no new categories emerged. Therefore, data collection was considered complete with 10 participants.

Each interview lasted 30 to 60 minutes. The interviews were held in a private room that allowed two people to converse calmly at a site with convenient participant access. All participants requested to be interviewed at the hospital where they worked, and the first author conducted their interviews accordingly. Using a semi-structured interview guide, questions were asked in the order shown in Table 1.

An interview guide was developed for this study; it was pilot tested with two nurses. After evaluating their responses, a few questions were revised. One of the modifications was adding "evidence"

**Table 1. Interview guide.**

| Questions | |
|---|---|
| 1 | Do you disinfect the patient's skin before subcutaneous injection? |
| 2 | Do you think that skin disinfection before a subcutaneous injection is necessary? |
| 3 | What is the evidence and reasoning behind these beliefs? |
| 4 | How do you perceive the reasons for skin disinfection in clinical practice? |
| 5 | What do you think about omitting skin disinfection? |

to the third question to elicit concrete evidence. A second modification was the inclusion of the fifth question, broadly asking about participants' ideas regarding omission of skin disinfection before subcutaneous injection. At the beginning of the interview, participants were asked about their years of experience as a nurse, their years of experience as a CNIC, and their department. They were also asked some follow-up questions such as "What do you mean?" to clarify some answers or "Can you explain further?" to encourage them to expound on their narrative. To ensure the consistency and accuracy of the data, the interviews were recorded with the participants' permission. The first author who conducted the interviews transcribed them verbatim, and prepared field notes during and immediately after the interviews. There were no repeat interviews.

## Data analysis

An inductive content analysis was used for our data analysis[20]. The researchers read each verbatim transcript several times to obtain an overall understanding of the content and gain a sense of the whole [21]. The meaning units in the interviews related to nurses' reasoning process were identified and coded. The codes were sorted into subcategories based on similarities and differences [22]. Depending on the relationships among subthemes, a larger number of subthemes can be organized, or combined, into a smaller number of themes [22]. After assessments across subcategories, overarching themes were derived. When discrepancies in coding occurred, the researchers of this study discussed and resolved them through consensus. The process was repeated until the content of each interview was compared with the content of all other interviews. Through the process, emerging findings could be identified and comparative commonalities could be extracted. This series of analyses methods were performed by three researchers. Two of them are nurses with experience of working in hospitals; the other is an occupational therapist with experience in medical practice. All three are experienced in qualitative research.

## Study rigor

Rigor was confirmed following Lincoln and Guba's criteria [23]. The credibility of the research findings was established using member checking and peer debriefing. Transferability was ensured via detailed descriptions of the research process. Dependability was achieved by checking the consistency of the findings. The first author, who conducted the interviews, did not have a prior relationship with the participants, which helped participants to freely provide their opinions and perceptions, which were accurately transcribed to promote authenticity.

## Ethical considerations

This study followed the guidelines set out by the 1975 Helsinki Declaration (2008 version). The study was approved by the Hokkaido University Graduate School of Health Sciences Ethics Committee and the ethics committee of the study site (13–51). When briefing potential participants, the researchers explained the purpose and requirements of the study, participants' right to withdraw at any time without consequences, and possibility of the authors publishing

the results. They were informed that personal information would be managed appropriately, and that colloquial and written data would be discarded at the end of the study. The above aspects were explained verbally and in writing, and written consent was obtained. After this, the researchers began to schedule interviews.

## Results

The researchers conducted semi-structured interviews with 10 CNIC, followed by an inductive content analysis. Seven themes emerged around the rationale for why CNIC performed skin disinfection before administering subcutaneous injections, namely, "common practice," "presumed merit of omitting disinfection," "adherence to hospital norms," "avoiding persistent suspicion of infection," "no detrimental effect," "ingrained custom," and, "tacit approval for not disinfecting in home care settings." The meaning of each theme is elaborated upon, using direct quotations from the participants in Table 2.

### Common practice

The decision to disinfect or not was influenced by perceptions and responses of the people who received care. One reason for skin disinfection before administering subcutaneous injections was that it is common practice. Participants worried that omitting this step would not be acceptable, and would instead induce anxiety in the patient. They also reported that they would agree to omit skin disinfection if it became a common practice with injection patients among the general public. Thus, disinfection was carried out owing to participants' perception that it is a current common practice.

### Presumed merit of omitting disinfection

Some merits of omitting skin disinfection before subcutaneous injection as part of standard care were mentioned. These included the economic benefits of reducing the cost of purchasing cotton for disinfection and disposing of waste, reduction in labor by skipping one of the steps involved in administering injections, and avoidance of unnecessary irritation to the skin caused by disinfectant solutions. Although these are small benefits, they can aid in elevating the patients' comfort level. As described above, the participants recognized the specific benefits of omitting skin disinfection before subcutaneous injection.

### Adherence to hospital norms

Hospitals have standards that staff must follow to provide patients with a certain quality of care. Participants said that, because they worked in a hospital, they followed hospital norms. Even if they personally believed that skin disinfection was unnecessary, the hospital rule was to disinfect the skin before administering every injection. Hence, they had no option to skip the step of skin disinfection. Thus, one of the reasons nurses used skin disinfection before subcutaneous injection was adherence to hospital norms.

### Avoiding persistent suspicion of infection

Participants were concerned about the risk of infection when skin disinfection was omitted before subcutaneous injection. They reported that the purpose of alcohol disinfection before administering subcutaneous injections was to remove bacteria from the skin and prevent infection. Disinfection may not completely prevent infection, but it is practiced on the assumption that the risk of infection can be reduced. Participants recognized that omission of skin disinfection before subcutaneous injection was unlikely to cause infection based on

**Table 2. Themes, subthemes, and quotations from the interviews.**

| Themes | Subthemes | Quotations |
|---|---|---|
| Common practice | Public perception of the need to disinfect before injection | • If the knowledge that there is no need to disinfect the skin before subcutaneous injection were to become widespread and accepted, I would consider omitting skin disinfection. |
| | | • It is common practice now to disinfect before an injection, so we do it to provide care to patients in this situation. |
| | | • I think the way of thinking will change a little if people are familiarized with the information that it is okay to not disinfect. |
| | Patient anxiety can occur by omitting disinfection | • If there is swelling that is caused by the local reaction of the injection, the patient may be worried that it may have been caused by not disinfecting. |
| | | •I do not know if patients are convinced or not about omitting skin disinfection. |
| | | · Omission of disinfection under current circumstances has psychological effects on patients. |
| Presumed merit of omitting disinfection | The certainty of the procedure is increased by simplifying the procedure if omitted | • The good thing for us is that the time taken for the treatment may be a little shorter. The reduction of one process can lead to operational efficiency. We will be able to focus on ensuring that patients' injections are administered. |
| | | • Because there is less to prepare, it becomes easier to do. |
| | Expected economic benefit if omitted | • Although garbage is a small issue, if we consider each patient individually, it becomes a big issue when considering many diabetes patients. |
| | | • Omitting this step can therefore help in cost reduction for the hospitals because the amount of antiseptic cotton purchased will be reduced. |
| | Decrease in harm to patients from alcohol | • There are many patients who are atopic. These patients have to tolerate skin pain from alcohol wipes. Most patients persevere by saying they are fine. |
| Adherence to hospital norms | Disinfection of skin in hospitals with no choice | • In the ordinary course of hospital work, the option for me or other healthcare professionals not to disinfect the patient is not up for debate. |
| | Adherence to the manual | • Disinfection is a standard nursing procedure in the hospital, and I believe it should be practiced as long as it remains so. |
| | | • I do not think disinfection will be done if the manual is revised. |
| | Difficulty to implement as CNIC | • Personally, I do not think skin disinfection before administering subcutaneous injection is necessary, but in my current position, I disregard my own opinion and follow the norm. |
| | | • I am responsible for infection control in the hospital, but I do not think there is any need to change the practice or omit disinfection under the current circumstances. |
| Avoiding persistent suspicion of infection | Skin disinfection to remove risk of infection as much as possible | • The reason to carry out skin disinfection is that it minimizes the risk of causing an infection in the patient. |
| | Insufficient convincing evidence | • If the CDC guidelines say it is unnecessary, then I will definitely believe it, because those guidelines are based on a considerable evidence. |
| | | • Previous foreign studies cannot be applied to Japanese people as they are. If there is evidence that Japanese people really do not have any trouble, I can do it. |
| | Difficulty in persuading CNIC to omit the practice of disinfection | • I thought that is one way to interpret it when I read the previous research that skin disinfection before subcutaneous injection is not always necessary. However that does not motivate me to change my behavior. |
| | | • In my head, I knew that the pH under the skin is a pH that does not allow bacteria to grow, so it does not lead to infection. |
| | Perceived awareness of cleanliness | • If it is not necessary to disinfect the skin, it is important to instruct the patient to keep the skin clean. However, at present, the patient has not been instructed to do so. |
| | | • If it is not necessary to disinfect skin before injection, awareness of cleanliness as a whole may be lowered, and washing hands may be neglected. |
| | Required ability to adequately determine whether disinfection is necessary | • I do not think the results of the previous study can be applied to patients with low immunity. |
| | | • I am concerned about whether patients can judge if they need disinfection or not. |

(*Continued*)

**Table 2.** (Continued)

| Themes | Subthemes | Quotations |
|---|---|---|
| No detrimental effect | The absence of significant patient disadvantage caused by disinfection | • To be honest, I do not hear much about disinfection being detrimental to the patient. |
| | | • Compared with other infection control issues, there are no major disadvantages for patients even if skin disinfection is continued. |
| | Minor problems with alcohol exposure | • The only harm to the patient is the redness of the skin caused by the accidental use of an alcohol swab to an alcohol-hypersensitive person. |
| Ingrained customs | Resistance to overturning a convention | • We have been in a situation where we have been disinfecting the skin before subcutaneous injections for a long time now. Hence, it is hard to teach everyone the rationale for why they do not have to do it in future. |
| | | • There is concern that nurses may be confused if the need for skin disinfection differs depending on the injection. |
| | The education nurses have received about the need to disinfect | • The reason skin disinfection is always carried out in the clinical setting is that we all learned it from nursing skills textbooks. That is why there is no doubt in anyone's mind. |
| Tacit approval for not disinfecting in home care settings | Acknowledgment of those who omit disinfection | • In the case of a person with dementia, I think the disinfection procedure may be forgotten before the self-injection of insulin. However, it is more important to inject insulin than to disinfect the skin. |
| | | • I believe it is okay to skip skin disinfection at home, because, unlike in a hospital, at home, the problem is restricted to just the person, and it becomes their responsibility. |
| | Unnecessary skin disinfection is a burden for patients at home | • I wonder if such a procedure is really necessary when the patient has to continue taking insulin at their home. |
| | | • In the case of people who inject subcutaneously on a daily basis, omitting disinfection reduces the burden on the subject. |

Note: CNIC refers to the Certified Nurses in Infection Control in Japan.

literature demonstrating that skin disinfection prior to subcutaneous injection was unnecessary and knowledge of subcutaneous anatomical physiology. However, they were still concerned about infection, and it was difficult for them to actually introduce the practice of omitting disinfection before subcutaneous injections. If omitting disinfection of the skin before subcutaneous injection became the standard, it was feared that disinfection might be omitted even in situations where disinfection was necessary. To avoid persistent suspicion of infection, the nurses continued the practice of skin disinfection before subcutaneous injection.

## No detrimental effect

One reason considered by the participants for continuing disinfection was that it posed no significant harm to the patient. Although problems owing to exposure to alcohol could occur, they were not considered a significant disadvantage compared with many other infection control issues in hospitals. Participants perceived that it was not necessary to actively consider omitting the practice of disinfection prior to subcutaneous injections, as it is not detrimental for patients if continued.

## Ingrained custom

The nurses were taught that disinfection before subcutaneous injection was necessary from the time they were students, and there was no opportunity to reflect upon the necessity of the practice even after they had started work. Nurses routinely administer injections after skin disinfection without questioning its scientific basis. Thus, it has become a deeply ingrained practice. They felt that it would be difficult to change this convention because it had become a custom.

Another reason for performing skin disinfection before subcutaneous injections was that it was perceived as an ingrained practice.

## Tacit approval for not disinfecting in home care settings

Participants believed that skin disinfection may not be necessary for patients at home because it places an extra burden on patients. When patients require injections in home care settings, it is important to ensure that the required dose is injected, and skipping skin disinfection is not considered a problem. In home care settings, nurses give tacit approval to the omission of disinfection because there are priorities over adhering to the norms of skin disinfection before subcutaneous injection.

## Reasoning for disinfecting before subcutaneous injection

Seven themes emerged from the data examined in this study. Fig 1 shows the reasoning of the CNIC for disinfection skin before administering subcutaneous injections. Patients expect skin to be disinfected before a subcutaneous injection because it is "common practice." Before administration, there was "presumed merit of omitting disinfection," such as reduced patient burden for self-injection and streamlining and efficiency for nurses. However, because disinfection does not harm the patient, nurses continued the precedent to avoid any possible risk of infection; this concept was categorized as "ingrained custom," "no precise effect," and "avoiding persistent suspicion of infection." In the hospital, CNIC, as hospital staff, must "adhere to hospital norms" and, thus, do not have the choice of omitting the practice of skin disinfection. Outside the hospital, on the contrary, CNIC are not obliged to follow these norms, and the need for skin disinfection was determined based on the priorities of individual self-injecting patients. Thus, there was "tacit approval for not disinfection in home care settings."

The seven themes identified in this study are enclosed in squares. "Common practice" includes six other themes because they were based on common practice. The decision on whether to disinfect the skin before subcutaneous injection was made by going back and forth between the four themes: a positive theme of "presented merit of omitting disinfection," and

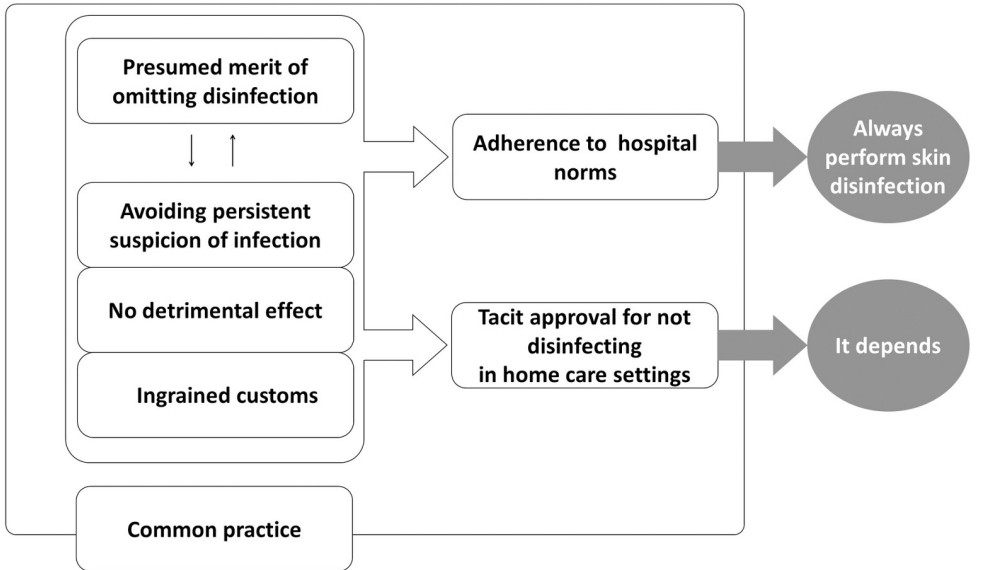

**Fig 1. Reasoning process for disinfecting skin before subcutaneous injection.**

three other negative themes on the omission of said practice. In considering them, the decision was made in the presence of patients and/or injection providers. In the case of a hospital, the decision went through "adherence to hospital norms," and then "always perform skin disinfection"; and if the patient was at home, it went through "tact approval for not disinfecting in home care settings," and finally, "it depends on the patient's situation."

## Discussion

The purpose of this study was to describe CNIC' reasoning for disinfecting or not disinfecting skin before administering a subcutaneous injection. A qualitative inductive content analysis generated seven themes. Although extant research of the past 50 years has consistently shown that swabbing the skin with alcohol before administering a subcutaneous injection is unnecessary [11, 24, 25], in clinical settings, nurses continue to disinfect skin, as shown in the theme of "ingrained custom." Thus, this practice can be described as based on tradition and habits.

The process of de-implementation is necessary to transform such a traditional and habitual practice. De-implementation is the process of identifying and removing practices based on tradition and habits that lack adequate scientific support [26]. In the pursuit of evidence-based health care, de-implementation of old routines is just as important as the implementation of new evidence [26]. In this study, "ingrained custom," "common practice," "adherence to hospital norms," "avoiding persistent suspicion of infection," and "no detrimental effect" were identified as barriers and resistance to the process of promoting de-implementation of skin disinfection before subcutaneous injections. It is expected that removing these barriers and resistances would result in updated evidence-based practices.

Notably, CNIC's reasoning regarding disinfection differed between hospitals and home care settings. In the hospital, a nurse is a staff member, and nursing services are provided in accordance with hospital standards. Therefore, it is not possible to deviate from hospital rules. Participants were aware of the negative effects of skin disinfection on patients, such as increased pain [27] and skin stiffness [28]. However, even if individual nurses judge skin disinfection as unnecessary in individual cases, they do not have the authority or choice to omit the practice of disinfection in a hospital setting.

In contrast, as described in the theme of "tacit approval for not disinfection in home care settings," CNIC opted to omit disinfection depending on the patient's circumstances in home care settings because the mandatory norms did not need to be strictly followed there. It was suggested that the process of de-implementation may already be underway, in part, in home care settings wherein subcutaneous injections are implemented. Sexson, Lindauer, and Harvath [29] reported that skin disinfection before administering subcutaneous injections is not necessary in a home care setting. In fact, in people with diabetes who routinely self-administer subcutaneous insulin injections, the skin disinfection rate was only 16% in Spain [30] and 30% in Italy [31], and no major problems have been reported. Intermittent urethral catheterization is one example of a difference in medical technique required in the hospital and in home care settings. It is often self-administered by patients at home, and the use of antiseptic solutions during insertion has been a subject of much debate [32]. Although the risk of a urinary tract infection is always present at urethral catheterization, many recent studies support the use of a clean, rather than sterile technique when patients insert intermittent urethral catheterization in the home environment [33]. Considering the psychomotor and psychological burdens of patients who self-inject, as well as the patient's family members who administer injections, there is no need to force disinfection before subcutaneous injection in the home environment. Nevertheless, introducing the decision to skip skin disinfection before subcutaneous injections is a concern in terms of infection. This concern can also be seen in the theme "avoid persistent suspicion of infection," which is one factor influencing the decision to

continue to disinfect the skin before administering subcutaneous injections. Skin commensal bacteria such as coagulase-negative staphylococci are major pathogens in the nosocomial setting [34]. An effective way to reduce the transmission of these health care-associated pathogens and the incidence of health care-associated infection is hand antisepsis [35]. When the omission of skin disinfection before subcutaneous injections is introduced to a patient, it is necessary to work even harder on hand hygiene to reduce concerns about infection among both nurses and patients.

Based on the results of this study and previous studies, evidence-based health care guidelines should be developed for skin disinfection before subcutaneous injections that consider the quality of life of patients at home, their physical conditions, and the surrounding environment at the time of injection.

In the UK, vaccine guidelines clearly state that disinfection is not required before administering vaccinations [36]. However, in Japan, official documents state that the skin must be disinfected before administering subcutaneous injections [37]. Karkos and Peters [38] and Schoonover [39] report that barriers to evidence-based practice that lack authoritative support ranked high against their introduction into clinical care. Even if nurses continue to update their knowledge and skills, the introduction of new ideas is difficult unless the authority of their facility/institution accepts those ideas.

Further, general nurses are trained to follow guidelines. Nursing education requires education for clinical nurses and basic nursing education. Parallelly, with the development of the new guidelines for home care settings, it is essential to develop the ability to evaluate the patient's skin condition from a multifaceted viewpoint from the stage of basic nursing education, and to emphasize the importance of not only following the guidelines, but also judging the necessity of skin disinfection in accordance with the individual patient's situation and instructing the patient in future nursing education.

## Limitations

The results of this study suggest that nurses have different reasoning for skin disinfection in home care settings than in hospitals. The participants in this study were experienced nurses who worked in hospitals. However, nurses who work in home care settings may have different reasoning on this issue. Further studies should consider the latter group to more deeply explore why nurses disinfect skin before administering subcutaneous injections. A quantitative survey will be required to clarify the actual status of skin disinfection before subcutaneous injections in home care settings when developing guidelines.

## Conclusion

Our study described CNIC's reasoning for disinfecting or not disinfecting skin before administering subcutaneous injections. Followed by an inductive content analysis, seven themes emerged: "common practice," "presumed merit of omitting disinfection," "adherence to hospital norms," "avoiding persistent suspicion of infection," "no detrimental effect," "ingrained custom," and, "tacit approval for not disinfecting in home care settings." Participants in this study acknowledged practicing home care subcutaneous injections without prior skin disinfection. Within hospitals, however, compliance with hospital norms, rather than judgment about individual patient conditions, prevails. This study reveals the barriers and resistance to promoting evidence-based practice in skin disinfection before subcutaneous injections at clinical settings. Overall, hospital norms had the most influence on CNIC's decision to disinfect.

## Author Contributions

**Conceptualization:** Yuko Yoshida, Rika Yano.

**Data curation:** Yuko Yoshida, Risa Takashima, Rika Yano.

**Formal analysis:** Yuko Yoshida, Risa Takashima, Rika Yano.

**Funding acquisition:** Yuko Yoshida.

**Investigation:** Yuko Yoshida.

**Methodology:** Yuko Yoshida, Risa Takashima, Rika Yano.

**Writing – original draft:** Yuko Yoshida, Risa Takashima, Rika Yano.

**Writing – review & editing:** Yuko Yoshida, Risa Takashima, Rika Yano.

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
