## [Decision Letter · Decision Letter 0]

23 Jul 2020

PONE-D-20-14697

Is skin disinfection before subcutaneous injection necessary? The reasoning of certified nurses in infection control in Japan

PLOS ONE

Dear Dr. Yoshida,

Thank you for submitting your manuscript to PLOS ONE. After careful consideration, we feel that it has merit but does not fully meet PLOS ONE’s publication criteria as it currently stands. Therefore, we invite you to submit a revised version of the manuscript that addresses the points raised during the review process.

We look forward to receiving your revised manuscript.

Kind regards,

Wen-Jun Tu

Academic Editor

PLOS ONE

Journal Requirements:

2.In your Data Availability statement, you have not specified where the minimal data set underlying the results described in your manuscript can be found. PLOS defines a study's minimal data set as the underlying data used to reach the conclusions drawn in the manuscript and any additional data required to replicate the reported study findings in their entirety. All PLOS journals require that the minimal data set be made fully available. For more information about our data policy, please see http://journals.plos.org/plosone/s/data-availability.

Reviewers' comments:

Reviewer's Responses to Questions

**Comments to the Author**

1. Is the manuscript technically sound, and do the data support the conclusions?

Reviewer #1: Yes

Reviewer #2: Partly

2. Has the statistical analysis been performed appropriately and rigorously? 

Reviewer #1: Yes

Reviewer #2: N/A

3. Have the authors made all data underlying the findings in their manuscript fully available?

Reviewer #1: Yes

Reviewer #2: No

4. Is the manuscript presented in an intelligible fashion and written in standard English?

Reviewer #1: Yes

Reviewer #2: Yes

5. Review Comments to the Author

Reviewer #1: Dear Authors,

Thank you for your well written manuscript. The chosen theme is important, connected to the context, traditions and cultures in different countries. Hygienical care of the skin during subcutaneous injections seems to be a discussion theme that has no end. However, I would like to ask you how the theme of hand hygiene is raised when the nurses choose to protect the skin with or without disinfection? How do the nurses discuss the consequenses of skin hygiene during subcutaneous injections with the patient? These could be mentioned in the discussion part.

According to the method: The study is qualitative. It could be more clear if the comparative method that you used is also mentioned in the abstract. You also use content analysis in the study. Is this right method to this study (together with the comparative method)? The manuscript could be more clear if there is for example a table where you present the results of the comparative part.

The figure is missing a short explanation what it presents.

Reviewer #2: The review is uploaded as an attachment

Methods and Materials / Analysis

There are major flaws in the method part that need to be processed and written in a clearer way for the reader, please develop methods section for a better dependability and confirmability. Its most important for this studies credibility.

I cant’t follow you step by step in the preparation, organizing and resulting phases in the content analysis process. Qualitative content analysis process/method is not clear described, what design have used according to your references?

The material needs to be further processed - seven themes and no subcategories?

The results section and the tables must also be presented more clearly, sometimes too many and long quotes -see even over that part.

The results of the discussion must be problematized and developed further

6. PLOS authors have the option to publish the peer review history of their article (what does this mean?). If published, this will include your full peer review and any attached files.

Reviewer #1: No

Reviewer #2: No

---

## [Author Response · Author response to Decision Letter 0]

5 Sep 2020

Sep 5, 2020

Dr. Wen-Jun Tu

Academic Editor

PLOS ONE

1160 Battery Street

Koshland Building East Suite 225

San Francisco、CA 94111 United States

Re: Manuscript ID: PONE-D-20-14697

Thank you very much for your e-mail and review of the manuscript (PONE-20-14697) that we sent on May 17, 2020. We would like to thank you and the two reviewers for providing constructive comments regarding the improvement of the original manuscript. All changes have been made in response to you and reviewers’ suggestions and itemized responses to the individual reviewer’s comments are also attached. In addition, the entire manuscript was revised to solve grammatical problems and improve readability through detailed English proofreading. These revisions are also colored in blue in the main document.

Reviewer #1:

Q1. The chosen theme is important, connected to the context, traditions and cultures in different countries. Hygienical care of the skin during subcutaneous injections seems to be a discussion theme that has no end. However, I would like to ask you how the theme of hand hygiene is raised when the nurses choose to protect the skin with or without disinfection? How do the nurses discuss the consequenses of skin hygiene during subcutaneous injections with the patient? These could be mentioned in the discussion part.

A1. We appreciate that helpful suggestion. As you advised, it is essential for both nurses and patients to maintain hand hygiene, whether disinfected or not prior to subcutaneous injection. In particular, after educating patients to ensure that they wash their hands for their own hand hygiene, nurses should assess the need for skin disinfection, explain the pros and cons of skin disinfection to patients, and make a decision after gaining their understanding. This point has been added to the revised draft. (P. 28-29 lines 418 – 428).

Q2. According to the method: The study is qualitative. It could be more clear if the comparative method that you used is also mentioned in the abstract. You also use content analysis in the study. Is this right method to this study (together with the comparative method)? The manuscript could be more clear if there is for example a table where you present the results of the comparative part. 

A2. The phrase “comparative method” was not appropriate. I appreciate that you pointed this out. We have revised this term and discussed the inductive analysis; we also updated the references. We also modified abstract. In order to express clearly how the themes were extracted, we created a table 2 that includes the quotations from the participants, subcategories, and themes (P. 11 lines 164 – 175 and table 2). 

Q３.The figure is missing a short explanation what it presents.

A３. A short explanation has been added to the figure’s legend. (P. 26-27 lines 379-388). 

Reviewer #2:

Q1. There are major flaws in the method part that need to be processed and written in a clearer way for the reader, please develop methods section for a better dependability and confirmability. Its most important for this studies credibility.

I cant’t follow you step by step in the preparation, organizing and resulting phases in the content analysis process. Qualitative content analysis process/method is not clear described, what design have used according to your references?

The material needs to be further processed - seven themes and no subcategories?

The results section and the tables must also be presented more clearly, sometimes too many and long quotes -see even over that part.

A1. Thank you for pointing that out. As you said, the description of how the data were analyzed was inadequate, so we have revised the text. We analyzed the data inductively. We have modified the description of the analysis method, using references to follow the "content analysis process.” In order to express clearly how the themes were extracted, we have created a table that includes the quotations from the participants, subcategories, and themes. We also removed quotations from the participants in the results section. The quotations are shown in shorthand in Table 2 (P. 11 lines 164 – 175 and tabe2). 

Q2. The results of the discussion must be problematized and developed further.

A2. In order to further develop the results of the discussion, we added the discussion as follows: 

(1)Skin hygiene and patient instruction including hand antisepsis (P. 28-29 lines 418 – 428). 

(2)About de-implementation (P. 30-31 lines 450-462).

(3)Nurse education related to the guidelines (P. 31 lines 463 – 467).

(4)The necessity for new guidelines at home care settings (P. 29 lines 428 – 431).

Academic Editor

Q1. Keywords: Please put the words in alphabetical order．

A1. We revised manuscript accordingly. (P. 4 line 41). 

Q2. Clarify the design, inductive design? 

A2. This study had an inductive design, so have I modified the text to reflect that. (P. 6 line 86). 

Q3. More distinct written about the inclusion and exclusions criteria.

A3. We have added the inclusion criteria. The selection criteria were as follows: (1) worked as a staff nurse with more than 10 years of experience (2) acquired the CNIC qualification. We did not describe the exclusion criteria because there was no exclusions criteria. (P. 7 lines 97 – 100). 

Q4. Convenience sampling ? pureposeful sampling?

A4. The data collection method of this research was convenience sampling, so we added that information. (P. 7 line 100). 

Q5. How did you do when you contacted the nurses for asking them to participate? E-mail? Letter ? Phone call? How many nurses did you ask?

A5. To contact the study candidates, at first, we contacted the director of nursing at the hospital where the CNIC worked by telephone and postal mail. The directors of a hospital’s nursing department introduced their hospital's CNIC to us. And we visited with ten CNIC and explained the purpose of this study and asked them for cooperation in the research. (P. 7 lines 100-109).

Q6. I want a flow chart showing how the participants were included in the study.

A6.The participant selection process was poorly documented. Therefore, we have described the selection process in detail. With this description, we decided that the flowchart was unnecessary. But if you disagree, I will respect your opinion and add a flowchart. 

We added the following text: “The first author contacted the director of nursing at the hospital in which CNIC worked. We asked ten director of a hospital nursing department with over 300 hospital beds to introduce us to the CNIC by telephone and letter. One of the directors of a hospital’s nursing department declined to cooperate because the significance of the study was not understood. Eight directors of a hospital’s nursing department agreed to participate and introduced their hospital's CNIC to the researchers one by one. One Nursing Manager introduced two CNIC from her hospital. There are usually only one or two CNIC in hospitals. We visited with ten CNIC and explained the purpose of this study to the m and sought their voluntary participation.” (P. 7 lines 100-109).

Q7. In-depth interviews ? You write that semi-structured I. was used.

A7. Yes you are right. I conducted semi-structured interviews, so I deleted the phrase “in-depth” and I have added “semi-structured.” (P. 9 line 134). 

Q8. References missing in text .

A8. We provided the missing references. (P. 11 lines 164-175). 

Q9. Explaine more about the interview situation. Whitch demographic variables did you include in the questionnaire?

A9. We added the interview situation. All participants requested to be interviewed at the hospital where they worked, so the researchers conducted their interviews accordingly (P. 9-10 lines 145-147). We also added that demographic variables were collected to at the beginning of the interview (P. 10 lines 153-155). 

Q10. Did you have any follow up questions? (To clarify some answers) like “Can you explain more?”, “How do you mean”? were asked. Did you ask the nurses the demographic questions before you started…

A10. Yes, we followed up questions were asked during the interview. Therefore, we added about the follow-up question to the new manuscript (P. 10 lines 155-157). 

Q11. Who did the transcribed verbatim. The authors ? A secretary? 

A11. It was done by the first author who was the interviewer of all the interviews. (P. 10 line 159).

Q12. Was the interview guide developed for this study?

A12. Yes, it was. So we added the following text: The interview guide was developed for this study (P.10 lines 148).

Q13. Make a nicer table –

A13. I remade Table 1. 

Q14. I cant’t follow you step by step in the preparation, organizing and resulting phases in the content analysis process. Qualitative content analysis process/method not clear described, what design have used according to your references? 

A14. We did not clearly describe process of content analysis in detail. The revised manuscript was described in detail with references as follow. 

“This study used an inductive content analysis [18]. The researchers read each verbatim transcript several times to obtain an overall understanding of the content and gain a sense of the whole [20]. The meaning units in the interviews related to nurses’ reasoning process were identified and coded. The codes were sorted into subcategories based on similarities and differences [19]. Depending on the relationships between subcategories, researchers can combine or organize them into larger headings [19].After assessing across subcategories, overarching themes were derived. When discrepancies in coding occurred, the investigators discussed and resolved them by consensus. The process was repeated until the content of each interview was compared with the content of all other interviews.” (P. 11 lines 164 – 175). 

Q15. I want a table presenting examples of the analysis process ;themes, subthemes, and quotations from the interviews. You must describe the process how these seven themes arise (dependability) It must be transparent. The material needs to be further processed - seven themes and no subcategories?

A15. In order to understand the process of analysis of how the seven themes were extracted, the themes, subcategories, and quotations from interviews are shown in Table 2. 

Q16. I prefer Demographic in the method section. In what kind of ward did the Participants work? How often did they administrate a subcutaneous injection ? seldom ? often ? education level?

A16. We have moved the description of the demographic to the methods section. And,　ｗe added the following in the method section: “One of the ten was a ward head nurse and one was a staff nurse. The other eight were not assigned to a ward but were assigned to a department that worked across the hospital as infection control specialists. The CNIC had all worked in several wards as staff nurses for more than ten years. Therefore, although there were participants who did not intervene frequently among co-workers, all participants were in a position to instruct nurses on infection control measures including skin disinfection before injection.” (P. 8 lines 112– 121). 

Q17. The results section and the tables must also be presented more clearly, sometimes too many and long quotes -see even over that part. develop the flow chart and present this in the resultsection.

A17. We created Table 2 in the Results section. The Interview quotations in the results section have been omitted from the main text of the paper and described in a concise form in the Interview quotations section of the table. 

Q18. I think you must discuss the importance of the nurses following guidelines or not and if you have any national guidelines to follow ?? in such cases how do you implement it? Suggestions for clinical implications?

A18. It’s a question of whether nurses follow the guidelines, we think they should follow the guidelines. However, this study found that nurses comply with the guidelines in hospitals, but not in home care environments. In other words, we thought it was necessary to discuss whether the guidelines in the hospital setting and the guidelines in the home care settings should be the same, or whether they need to be developed separately. This point has been added to the revised draft. (P. 31 lines 436 – 467; P. 29 lines 428 – 431). 

Q19. In the ethical section discussion, were the participates informed that they could withdraw from the study at any time ?

A19. I did explain to the participants that they could decline to participate in this study at any time, and I clarified this in the ethical considerations section. (P. 8 lines 124 – 125). 

Q20. What is your bullet points? What is new? knowledge gap ?

A20. What I would like to highlight the most was the following sentence in the first paragraph of the discussion. “Our study reviewed that CNIC's regarding registering disinfection different between hospitals and home care settings.” I have revised the first paragraph so that the reader can understand it. (P. 27 lines 398 – 400). 

Q21. I think your results indicate a very important situation – however you must discuss your old references, today a very high figure with multi drug resistance exists when we discuss hospital and hospital care. Coagulase-negative staphylococci (CoNS) are the most abundant micro-organisms colonizing human skin and mucous membranes all been shown to alter the flora, reduce the variety of strains and result in accumulation of multi-drug-resistant CoNS, I think this can be discussed more, I’m not convinced that the same guidelines should prevail in hospitals as well as in homes and at homecaresttings. The conditions are completely different.. This is a new important point to discuss.

A21. Thank you for your very constructive advice. Based on the results of the seven themes of this study and the results of previous studies, we thought that a home version of the guidelines should be developed. It should take into account the patient's physical condition and the environment at the time of injection, in addition to the guidelines applied in hospitals. In a previous study, the participants were outpatients, not hospitalized patients. At present, it has been pointed out that it is unclear whether omitting skin disinfection before subcutaneous injection is applicable to all patients. I mentioned this point in the 6th paragraph of the discussion. (PP. 28-29 lines 418 – 431). 

Q22. Discuss the nurse education in relation to guidelines?

A22. We believe that there is a need for education regarding the description of the guidelines for disinfection before subcutaneous injection. However, in addition to this, it is necessary to introduce evidence on the necessity of skin disinfection and to educate the patients on the merits and demerits of disinfection and the necessity of individual assessment. These aspects have been added to the manuscript. (P. 31 lines 463 – 467). 

Q23. Discuss de-implementation - - it is very hard to change a behavior.. de-implementation is a process for identifying and removing practices based on tradition and habits which lack adequate scientific support. In the pursuit of evidence-based health care, de-implementation of old routines is just as important as the implementation of new evidence. (Upvall MJ, Bourgault AM. De-implementation: a concept analysis. Nurs Forum. 2018.) https://doi.org/10.1111/nuf.12256. was shown for the commonly used antibiotics.

A23. Thank you for your advice on de-implementation. I read the paper you recommended. Our results clearly demonstrate the barrier and resistance of de-implementation in skin disinfection prior to subcutaneous injection. We also thought it was important to educate nurses in order to advance the de-implementation process. These are added to the discussion section. (P. 30-31 lines 450 – 467 ). 

Q24. Try to make the conclusion more spot on, I think it is too long.

A24. We shortened the conclusion. (P. 32-33 lines 486 – 499). 

We have also modified the Data Availability statement.

We believe that we have addressed comments of editor and reviewers and hope that the revised manuscript is now acceptable publication in PLOS ONE. Thank you for your generous consideration.

Sincerely yours,

Yuko Yoshida

Hokkaido University, Faculty of Health Sciences, 

Department of Fundamental Nursing

Kita 12 Nishi5, Kita-ku, Sapporo, Hokkaido 060-0812, Japan

Phone and FAX: +81-(0)11-706-3718

Email address: yuko790402@hs.hokudai.ac.jp

---

## [Decision Letter · Decision Letter 1]

16 Sep 2020

PONE-D-20-14697R1

Is skin disinfection before subcutaneous injection necessary? The reasoning of Certified Nurses in Infection Control in Japan

PLOS ONE

Dear Dr. Yoshida,

Thank you for submitting your manuscript to PLOS ONE. After careful consideration, we feel that it has merit but does not fully meet PLOS ONE’s publication criteria as it currently stands. Therefore, we invite you to submit a revised version of the manuscript that addresses the points raised during the review process.

We look forward to receiving your revised manuscript.

Kind regards,

Wen-Jun Tu

Academic Editor

PLOS ONE

Reviewers' comments:

Reviewer's Responses to Questions

**Comments to the Author**

1. If the authors have adequately addressed your comments raised in a previous round of review and you feel that this manuscript is now acceptable for publication, you may indicate that here to bypass the “Comments to the Author” section, enter your conflict of interest statement in the “Confidential to Editor” section, and submit your "Accept" recommendation.

Reviewer #1: All comments have been addressed

2. Is the manuscript technically sound, and do the data support the conclusions?

Reviewer #1: Partly

3. Has the statistical analysis been performed appropriately and rigorously? 

Reviewer #1: Yes

4. Have the authors made all data underlying the findings in their manuscript fully available?

Reviewer #1: Yes

5. Is the manuscript presented in an intelligible fashion and written in standard English?

Reviewer #1: Yes

6. Review Comments to the Author

Reviewer #1: Dear Authors,

Thank you for the possibility to review this paper again. I can see that a lot of work have been done, but I feel there is lot to still do. I wonder if the whole research group participated to R1 or was it only the first author of this paper?

Abstract: Start the A. by going straigt, with 1-2 sentences, to the corn substance, not to previous litterature. Include: The study was done in autumn 2013. Please revise the A. The study does not "explain" you need to use "describe". Same words that you have in the text. A. needs much more stringency. "However" is not a good word in the abstact. Keywords: several missing: Japan, s.c injections (not only inj), content analysis. "Bake in Qualitative" into the A.

Introduction: Use same words throughout the whole paper. Now you use respondent, informant and participant at the same time, also interviewer and researcher. This is confusing. Reduce the use of the word "participant" and "therefore" (in the whole paper).

What is the main research question in this study? This should be clearly written.

Methods: Datacollection (DC). What about having a separate part for the study design and putting together DC and participants as a overhead. Now this text is mixed. Generally now its more clear. I think that you do not need a flow chart according to the participants.

Ethical considerations (E): This part should be moved before the results part. Now it confuses the reader in the method part. Keep it as a separate part for to make the paper clear. You need references to the E. since human beings participated. Use for example Helsinki Declaration and some local ethical committee descriptions from Japan. This is important.

Line 103-104 p. 6 is unclear.

Line 107, p. 6: ...were willing to participate in the survey... Do you mean in the study or what?

Line 135-137. p. 8. This is irrelevant info. Please remove curriculum vitae information from the manuscript.

Results: Where did the results come from? Please shortly introduce this in the ingress. Did the previous studies influence the results? Or the interviews with participants? What about your method?

The results part has in general stepped backwords and it feels that the you have lost the abstraction level. The text is now repeating the figure and the table. Please go back to your original manuscript and discuss the presentation of the results in your research group. This needs stringency and revision.

Discussion: This part includes also many repeatings. Here you should raise the level of abstaction even more and discuss the paper in whole. Please discuss this part with your reseach group and revise the text.

Line 365, p. 23. Is this necessary to mention? Many readers know this.

Line 382, p 23. NB, the English Language.

In general: I recommend you to use a professional English Language control.

7. PLOS authors have the option to publish the peer review history of their article (what does this mean?). If published, this will include your full peer review and any attached files.

Reviewer #1: No

---

## [Author Response · Author response to Decision Letter 1]

31 Oct 2020

Oct 31, 2020

Dr. Wen-Jun Tu

Academic Editor

PLOS ONE

Re: Manuscript ID: PONE-D-20-14697R1

Dear Editor:

Thank you very much for your e-mail and review of the manuscript (PONE-20-14697R1) that we sent on Sep 5, 2020. We are very grateful for your constructive and insightful comments regarding the improvement of the revised manuscript. All changes have been made in response to your and the reviewers’ suggestions and itemized responses to the individual reviewer’s comments are also attached herewith. In addition, the entire manuscript was revised to resolve grammatical issues and improve readability through detailed proofreading. These revisions are also highlighted in red in the main document.

I look forward to working with you and the reviewers to move this manuscript closer to publication in PLOS ONE. Thank-you for your consideration. I look forward to hearing from you.

Sincerely,

Yuko Yoshida

Hokkaido University Faculty of Health Sciences

Address: Kita 12 Nishi 5, Kita-ku, Sapporo, Hokkaido 060-0812, Japan 

Tel.:+81-011-706-3718

Fax: +81-011-706-3718

Email: yuko790402@hs.hokudai.ac.jp

Question1:

Abstract: Start the A. by going straigt, with 1-2 sentences, to the corn substance, not to previous litterature. Include: The study was done in autumn 2013. Please revise the A. The study does not "explain" you need to use "describe". Same words that you have in the text. A. needs much more stringency. "However" is not a good word in the abstact. 

Response1:

We modified the sentence in the first line, deleted the section on previous literature, and added the following: “even though there is insufficient evidence on the need for disinfection.” We added that “interviews were conducted with 10 CNIC in 2013.” We also corrected and used the term “describe” instead of “explain,” Additionally, we reviewed the entire paper and avoided the use of the word “explain.” We have changed the expression from “however” to “on the contrary” in the abstract. We have made the above modifications to improve the stringency of abstract.

Question2:

Keywords: several missing: Japan, s.c injections (not only inj), content analysis. "Bake in Qualitative" into the A.

Response2:

We added “Japan,” “subcutaneous injections,” “content analysis” as keywords. We were not sure what the phrase “Bake in Qualitative” into the A. means. Thus, we have not been able to correct for this. We are sorry to trouble you; we request you to kindly explain what you meant further. We will correct this as soon as you let us know.

Question3:

Introduction: Use same words throughout the whole paper. Now you use respondent, informant and participant at the same time, also interviewer and researcher. This is confusing. Reduce the use of the word "participant" and "therefore" (in the whole paper).

What is the main research question in this study? This should be clearly written.

Response3:

We apologize for the confusion caused by the use of “respondent,” “informant,” and “participant” interchangeably; we have corrected them all to “participant.” In addition, the term “interviewer” has been changed to “researcher” to make it easier to follow. We reduced the usage of “participant” and “therefore” throughout the paper (participant:41→38, therefore:11→6). Special attention was paid to the use of unified terms and selection of conjunctions while proofreading.

We have modified our research question in the introduction as follows. “The research questions in this study were why Japanese nurses are performing skin disinfection before subcutaneous injections and what is the background to the practice. Based on this research question, the purpose of this study was to describe nurses’ reasoning for disinfecting or not disinfecting the skin before administering a subcutaneous injection, using a qualitative design.” (P. 4 lines 78-82).

Question4:

Methods: Datacollection (DC). What about having a separate part for the study design and putting together DC and participants as a overhead. Now this text is mixed. Generally now its more clear. I think that you do not need a flow chart according to the participants.

Response4:

Thank you for your advice. We outlined the “study design” in one section and then delineated the “participants and data collection” section together.

Question5:

Ethical considerations (E): This part should be moved before the results part. Now it confuses the reader in the method part. Keep it as a separate part for to make the paper clear. You need references to the E. since human beings participated. Use for example Helsinki Declaration and some local ethical committee descriptions from Japan. This is important.

Response5:

We removed the ethical considerations section before the Results part and have added it as a separate section. We added the following sentence: “This study followed the guidelines set out by the 1975 Helsinki Declaration (2008 version). The study was approved by the Hokkaido University Graduate School of Health Sciences Ethics Committee and the ethics committee of the study site (13-51).” (P. 9 lines 177-185).

Question6:

Line 103-104 p. 6 is unclear.

Response6:

We believe our intended meaning was not conveyed due to a spelling error. We have corrected it; kindly let us know if you seek further clarity. 

(Before) We visited with ten CNIC and explained the purpose of this study to the m and sought their voluntary participation.

(After) The researchers visited the 10 CNIC, explained the purpose of the study to them, and sought their voluntary participation. (P. 5 lines 110-P.6 lines111).

Question7:

Line 107, p. 6: ...were willing to participate in the survey... Do you mean in the study or what?

Response7:

We apologize for the confusion; we have revised “survey” to “study.”

(Before) Ten CNIC were willing to participate in the survey.

(After) The 10 CNIC agreed to participate in the study. (P. 6 line 144).

Question8:

Line 135-137. p. 8. This is irrelevant info. Please remove curriculum vitae information from the manuscript.

Response8:

We have deleted the section in adherence with your comment.

Question9:

Results: Where did the results come from? Please shortly introduce this in the ingress. Did the previous studies influence the results? Or the interviews with participants? What about your method?

Response9:

We added the information you require in the beginning of the Results section. 

“The researchers conducted semi-structured interviews with 10 CNIC and performed an inductive content analysis. Seven themes emerged around the rationale of why CNIC performed skin disinfection before administering subcutaneous injections: ‘common practice,’ ‘adherence to hospital norms,’ ‘ingrained custom,’ ‘no detrimental effect,’ ‘presumed merit of omitting disinfection,’ ‘avoiding persistent suspicion of infection,’ and ‘lowered adherence to norms in home care settings.’ The meaning of each theme is elaborated upon, using direct quotations from the participants in Table 2.” (P. 9 lines 188-194).

Question10:

The results part has in general stepped backwords and it feels that the you have lost the abstraction level. The text is now repeating the figure and the table. Please go back to your original manuscript and discuss the presentation of the results in your research group. This needs stringency and revision.

Response10:

We have tried to use subcategory-based descriptions for each theme’s description to provide a convincing explanation of how the theme was generated. As a result, we believe the contents of the table were repeated many times and the abstraction level was lost. We discussed this again in the research group and in the main text; we have modified the main text to describe the essential meaning of each theme, avoiding simple repetition of the table contents as much as possible. In addition, for the item “Reasons to perform skin disinfection before subcutaneous injection,” the number of repeats of legend in Fig. 1 increased. We have made revisions to minimize the duplication of descriptions between the two.

Question11:

Discussion: This part includes also many repeatings. Here you should raise the level of abstaction even more and discuss the paper in whole. Please discuss this part with your reseach group and revise the text.

Response11:

We deleted the repetition of the sentence. To increase the level of abstraction, we rethought the structure of the discussion section and rewrote it. The content of the structure is as follows: (1) summary of the results of this study; (2) comparison of the concept of de-implementation with the results of this study; (3) reference to the de-implementation of skin disinfection observed in some cases at home; (4) justification of the de-implementation of skin disinfection at home; and (5) development of guidelines to facilitate the de-implementation process of skin disinfection before subcutaneous injections.

Question12:

Line 365, p. 23. Is this necessary to mention? Many readers know this.

Response12:

We deleted the points you deem unnecessary.

Question13:

Line 382, p 23. NB, the English Language.

Response13:

We deleted the points you have mentioned.

Question14:

In general: I recommend you to use a professional English Language control.

Response 14:

We shared your comments with a publication support services company and asked for an experienced proofreader to check the language.

---

## [Decision Letter · Decision Letter 2]

16 Nov 2020

PONE-D-20-14697R2

Is skin disinfection before subcutaneous injection necessary? The reasoning of Certified Nurses in Infection Control in Japan

PLOS ONE

Dear Dr. Yoshida,

Thank you for submitting your manuscript to PLOS ONE. After careful consideration, we feel that it has merit but does not fully meet PLOS ONE’s publication criteria as it currently stands. Therefore, we invite you to submit a revised version of the manuscript that addresses the points raised during the review process.

We look forward to receiving your revised manuscript.

Kind regards,

Wen-Jun Tu

Academic Editor

PLOS ONE

Reviewers' comments:

Reviewer's Responses to Questions

**Comments to the Author**

1. If the authors have adequately addressed your comments raised in a previous round of review and you feel that this manuscript is now acceptable for publication, you may indicate that here to bypass the “Comments to the Author” section, enter your conflict of interest statement in the “Confidential to Editor” section, and submit your "Accept" recommendation.

Reviewer #1: All comments have been addressed

2. Is the manuscript technically sound, and do the data support the conclusions?

Reviewer #1: Partly

3. Has the statistical analysis been performed appropriately and rigorously? 

Reviewer #1: N/A

4. Have the authors made all data underlying the findings in their manuscript fully available?

Reviewer #1: Yes

5. Is the manuscript presented in an intelligible fashion and written in standard English?

Reviewer #1: Yes

6. Review Comments to the Author

Reviewer #1: Dear Author(s),

Thank you once again for the possibility to take part in your manuscript. I can see that a lot of work is done and that the manus is now much better. However, I would like you to think over some important things and review these. There har been challenges to understand your track changes and two type of colouring the manuscript (red and blue) and what these mean and also the underlining in tables. There are a few grammatical problems and also words that need to be written correctly. If you could do the suggested changes and present the manus without track changes it would be easier to follow your thoughts.

Abstract (A): The first sentence is a result. I suggest that you remove the words "continue" and "even though". Could "evidence-based" research be possible to use to highlight research?

Read carefully the A once more and see if you could higher the abstraction level some more.

Introduction (I): Research question(s) are a bit unclear. You name two questions and write about one in same sentence. How many questions do you have in this study? This is very important that you write it/them clearly. I miss also the study context you are using. This could be a part of the research question(s). Use rather "is" than "were" in line 85. p 5.

Line 58. "Some studies..." You need a better start for the sentence. "Some" is coming also in next sentence.

Line 79. Please name the author.

Line 96 p.6. What is the difference between perception and experience? This could be explained in the indroduction (with references).

Lines 101-108 are a part of introduction.

Go straight to the participants under the heading on line 100.

Line 188: What do you mean with "larger headings"?

Line 190: Who were the "investigators"? Do you mean researchers of this study?

Ethical Considerations (EC): This part is much better now.

Line 214: "After this..." sentence is unnecessary here. Write 1-2 short sentencences in the end of EC about publishing the results, since you are now planning to publish this study. What did you explain about publishing to the participants?

Table 2. I advice you to think about to use the terms themes and subthemes or categories and sub-categories instead of mixing these with each other. It has been challenging to see the table through track changes.

Results (R): Line 307 p. 17. Are you sure this is a right heading here? I suggest to form this and think about the sub-theme here once again. So many track changes here has given challenges to understand the results.

Discussion (D): Start your discussion with your study objective to lead the readers back to your corn substance. I dont understand why the example of urinary tract infection and catheterization is here, line 400 p.26.

Lines 405-414 are marked with blue colour. Is there a special reason for this?

In the D: you should mention clearly your future visions about your study substance, what to do next?

Limitations: you could see a bit more for this and also the conclusion.

Good luck with your manuscript!

Best wishes from the reviewer 1.

7. PLOS authors have the option to publish the peer review history of their article (what does this mean?). If published, this will include your full peer review and any attached files.

Reviewer #1: No

---

## [Author Response · Author response to Decision Letter 2]

13 Dec 2020

Dec 13, 2020

Dr. Wen-Jun Tu

Academic Editor

PLOS ONE

Re: Manuscript ID: PONE-D-20-14697R2

Dear Editor:

Thank you for your e-mail and review of the manuscript (PONE-20-14697R2) that we sent on Oct 31, 2020. We are very grateful for your constructive and insightful comments regarding the improvement of the revised manuscript. All changes have been made in response to the reviewers’ suggestions and itemized responses to the reviewer’s comments are also attached herewith. In addition, the entire manuscript was revised to resolve grammatical issues and improve readability through detailed proofreading. Additions are marked in red, and deleted text is tracked via the ‘Track Changes’ function in Microsoft Word. The line numbers in this document are based on the main manuscript with the revision history and revisions shown in-line. These revisions are also highlighted in red in the main document.

I look forward to working with you and the reviewers to move this manuscript closer to publication in PLOS ONE. Thank you for your consideration. I look forward to hearing from you.

Sincerely,

Yuko Yoshida

Hokkaido University Faculty of Health Sciences

Address: Kita 12 Nishi 5, Kita-ku, Sapporo, Hokkaido 060-0812, Japan 

Tel.:+81-011-706-3718

Fax: +81-011-706-3718

Email: yuko790402@hs.hokudai.ac.jp

 

Q1:There har been challenges to understand your track changes and two type of colouring the manuscript (red and blue) and what these mean and also the underlining in tables. There are a few grammatical problems and also words that need to be written correctly. If you could do the suggested changes and present the manus without track changes it would be easier to follow your thoughts.

A1: We would like to apologize for this confusion. We have improved the readability of the revised manuscript by unifying the color and using the track changes function in Word.

Q2:Abstract (A): The first sentence is a result. I suggest that you remove the words "continue" and "even though". Could "evidence-based" research be possible to use to highlight research?

Read carefully the A once more and see if you could higher the abstraction level some more.

A2: Thank you for your advice. I removed “continue” and “even enough” following your advice. To improve the abstract, we have modified the research background to highlight the significance of this research using the term “evidence-based.” The corrections and additions are in red.

Line 19-25

“Nurses continue to disinfect the skin before administering subcutaneous injections as a standard process in clinical settings; despite evidence that disinfection is not necessary. To implement evidence-based practice, it is critical to explore why this gap between “evidence” and “practice” exists. This study aimed to describe the reasons offered by Certified Nurses in Infection Control (CNIC) in Japan for performing skin disinfection before subcutaneous injection. Adopting an inductive qualitative design, interviews were conducted with 10 CNIC in 2013.”

In addition, the discussion section has been modified to describe the interpretation of the results using expressions that correspond to the expression of the research background. 

Line 29-38

“The themes (c) and (g) were cited as the main reasons affecting decision-making. The CNIC administered injections following skin disinfection in hospitals in accordance with hospital norms. On the contrary, outside the hospital, they administered subcutaneous injections without skin disinfection. All themes except (b) and (g) reflect the barriers and resistance to omitting skin disinfection, while (g) shows that it is already partly implemented in home care settings. It is necessary to create a guideline for skin disinfection before subcutaneous injection that considers the quality of life of patients at home, their physical conditions, and the surrounding environment at the time of injection, in addition to the guidelines applicable in hospitals.”

Q3:Introduction (I): Research question(s) are a bit unclear. You name two questions and write about one in same sentence. How many questions do you have in this study? This is very important that you write it/them clearly. I miss also the study context you are using. This could be a part of the research question(s). Use rather "is" than "were" in line 85. p 5.

A3: Regarding the research question, we mentioned two in our previous manuscript: “Why Japanese nurses disinfect the skin before administering subsidiary injections and what is the background to this practice,” but deleted “what is the background to this practice,” and unified research question. The corrections and additions are in red. 

Line 95-96

“Thus, we pose the following research question: why do Japanese nurses disinfect skin before administering subcutaneous injections?”

For the study context, we rewrote the following text before the text of the research question: 

Line 91-95

“Although omitting skin disinfection before administering subcutaneous injections is safe, reduces the burden on patients, and likely reduces costs, it has still not become standard clinical practice. Exploring the reasons for this lack of adoption in clinical practice in Japan will help us address obstacles to the introduction of new evidence.”

Q4:Line 58. "Some studies..." You need a better start for the sentence. "Some" is coming also in next sentence.

A4: We have modified the sentence as per your suggestion, and now included a direct in-text citation as the beginning of the paragraph.

Lines 54-55

“Dunleavy et al. and Hope, Hickman, Parry, and Ncube [7,8] report that not using an alcohol swab is a risk factor for skin and soft tissue infections.” 

Q5:Line 79. Please name the author.

A5: Following your advice, we have included the names of researchers. 

Lines 74-76

“A study conducted in Greece　by Theofanidis [13] indicates that nurses disinfect the skin before insulin injections as a longstanding medical ritual, although there is insufficient evidence on the need for disinfection.”

Q6:Line 96 p.6. What is the difference between perception and experience? This could be explained in the indroduction (with references).

A6: Thank you for your advice. The difference between “experience” and “perception” is now described in the introduction section, with quotations. Additionally, we included in the introduction section the reason for targeting CNIC’s “experience” and “perception.” The corrections and additions are in red.

Lines 77-99

“Nurses who do not have extensive knowledge of infection control may assume that skin disinfection before administration of subcutaneous injections prevents infection. According to recent books published in Japan on nursing techniques and skills, disinfection is necessary, while only a few books have introduced studies verifying that it is unnecessary [15]. In contrast, Certified Nurses in Infection Control (CNIC) receive certification from the Japan Nursing Association for specializing in infection control and having advanced nursing skills. CNIC have more experience, skill, and knowledge-based perceptions than other nurses regarding skin disinfection before subcutaneous injection. In this study, the word “experience” is defined as “practical knowledge, skill, or practice derived from direct observation of or participation in events or in a particular activity” [16], while the word “perception” is defined as “The way in which something is regarded, understood, or interpreted” [17]. Describing CNIC’s experience and perception of skin disinfection before subcutaneous injection was considered suitable for this objective. 

Although omitting skin disinfection before administering subcutaneous injections is safe, reduces the burden on patients, and likely reduces costs, it has still not become standard clinical practice. Exploring the reasons for this lack of adoption in clinical practice in Japan will help us address obstacles to the introduction of new evidence. Thus, we pose the following research question: why do Japanese nurses disinfect skin before administering subcutaneous injections? Based on this research question, the purpose of this study was to describe, using a qualitative design, CNIC’s reasoning for disinfecting or not disinfecting the skin before administering a subcutaneous injection.”

Q7:Lines 101-108 are a part of introduction.

Go straight to the participants under the heading on line 100.

A7: Thank you for your comments. Lines 101–108 has been moved to the introduction section. 

Lines 111-113

“To meet conditions similar to the skills required for a subcutaneous injection, the participant selection criteria were as follows: (1) worked as a staff nurse with more than 10 years of experience and (2) acquired CNIC qualification.”

Q8:Line 188: What do you mean with “larger headings”?

A8: We apologize for this expression. We have revised this sentence as follows.

Line 168-170

“Depending on the relationships among subthemes, a larger number of subthemes can be organized, or combined, into a smaller number of themes [22].”

Q9:Line 190: Who were the "investigators"? Do you mean researchers of this study?

A9: We have revised “investigators” to “researchers of this study.” 

Lines 171-172

“When discrepancies in coding occurred, the researchers of this study discussed and resolved them through consensus.”

Q10:Ethical Considerations (EC): This part is much better now.

Line 214: "After this..." sentence is unnecessary here. Write 1-2 short sentencences in the end of EC about publishing the results, since you are now planning to publish this study. What did you explain about publishing to the participants?

A10: Thank you for your comments. Following your advice, we deleted the sentence, and added more explanation on publishing the results to the participants. 

Lines 192-194

“When briefing potential participants, the researchers explained the purpose and requirements of the study, participants’ right to withdraw at any time without consequences, and possibility of the authors publishing the results.”

Q11:Table 2. I advice you to think about to use the terms themes and subthemes or categories and sub-categories instead of mixing these with each other. It has been challenging to see the table through track changes.

A11: We apologize for the confusion. We attribute this issue to using Excel. The manuscript submitted the time before last one was made in Excel. Therefore, it was created by using Word from the previous manuscript. We have corrected the terminology and made all the tables more readable.

Q12:Results (R): Line 307 p. 17. Are you sure this is a right heading here? I suggest to form this and think about the sub-theme here once again. So many track changes here has given challenges to understand the results.

A12: Thank you for your comments. We have reconsidered the subtheme of “lowered adherence to norms in home care settings,” since it consists of subthemes “acknowledgment of those who omit disinfection” and “unnecessary skin disinfection is a burden for patients at home.” We believe “tacit approval for not disinfecting in home care settings” would be a suitable theme. This is because nurses in hospitals never think about complying with the norm; in situations where the norm in home care settings is lax, they start to consider other priorities, and it can be interpreted that they are hesitant and even partially acquiescing in omitting skin disinfection. This has been revised as follows.

“Tacit approval for not disinfecting in home care settings”

Q13:Discussion (D): Start your discussion with your study objective to lead the readers back to your corn substance. I dont understand why the example of urinary tract infection and catheterization is here, line 400 p.26.

A13: Thank you for your advice. We have revised the section as follows.

Lines 304-309

“The purpose of this study was to describe certified nurses’ reasoning for disinfecting or not disinfecting skin before administering a subcutaneous injection. A qualitative inductive content analysis generated seven themes. Although extant research of the past 50 years has consistently shown that swabbing the skin with alcohol before administering a subcutaneous injection is unnecessary [11, 24, 25], in clinical settings, nurses continue to disinfect skin, as shown in the theme of “ingrained custom.”

The reasons given for the example of urinary tract infection and catheterization are as follows. Intermittent urethral catheterization requires a strictly sterile technique in the hospital, but not at home. This issue is hotly debated. In case of skin disinfection before subcutaneous injection, sterile technique is only required in hospitals, but this process can be omitted at home. There is always a risk of infection if sterile techniques are not used; even so, sterile techniques are not required when intermittent urethral catheterization is performed at home. The example of urinary tract infection and catheterization was presented as a good example as a result. We have revised the description below to convey our intention.”

Lines 338-344

“Intermittent urethral catheterization is one example of a difference in medical technique required in the hospital and in home care settings. It is often self-administered by patients at home, and the use of antiseptic solutions during insertion has been a subject of much debate [32]. Although the risk of a urinary tract infection is always present at urethral catheterization, many recent studies support the use of a clean, rather than sterile technique when patients insert intermittent urethral catheterization in the home environment [33].”

Q14:Lines 405-414 are marked with blue colour. Is there a special reason for this?

A14: We apologize for the confusion. There was no difference in the message depending on the color. This mistake has been corrected.

Q15:In the D: you should mention clearly your future visions about your study substance, what to do next?

A15: Thank you for this input. We have revised the implications and future scope of the study as follows. 

Lines 358-361

“Based on the results of this study and previous studies, evidence-based health care guidelines should be developed for skin disinfection before subcutaneous injections that consider the quality of life of patients at home, their physical conditions, and the surrounding environment at the time of injection.”

Q16:Limitations: you could see a bit more for this and also the conclusion.

A16: Thank you for your advice. The revised manuscript now includes the limitations, as shown below. 

Lines 383-390

“The results of this study suggest that nurses have different reasoning for skin disinfection in home care settings than in hospitals. The participants in this study were experienced nurses who worked in hospitals. However, nurses who work in home care settings may have different reasoning on this issue. Further studies should consider the latter group to more deeply explore why nurses disinfect skin before administering subcutaneous injections. A quantitative survey will be required to clarify the actual status of skin disinfection before subcutaneous injections in home care settings when developing guidelines.”

We accordingly modified our conclusion as well:

Lines 388-395

“Our study described CNIC’s reasoning for disinfecting or not disinfecting skin before administering subcutaneous injections. Followed by an inductive content analysis, seven themes emerged: “common practice,” “presumed merit of omitting disinfection,” “adherence to hospital norms,” “avoiding persistent suspicion of infection,” “no detrimental effect,” “ingrained custom,” and, “tacit approval for not disinfecting in home care settings.” Participants in this study acknowledged practicing home care subcutaneous injections without prior skin disinfection. Within hospitals, however, compliance with hospital norms, rather than judgment about individual patient conditions, prevails. This study reveals the barriers and resistance to promoting evidence-based practice in skin disinfection before subcutaneous injections at clinical settings. Overall, hospital norms had the most influence on CNIC’s decision to disinfect.”

In addition, the entire manuscript was revised to resolve grammatical issues and improve readability through detailed English proofreading.

---

## [Editor Report · Decision Letter 3]

26 Dec 2020

Is skin disinfection before subcutaneous injection necessary? The reasoning of Certified Nurses in Infection Control in Japan

PONE-D-20-14697R3

Dear Dr. Yoshida,

We’re pleased to inform you that your manuscript has been judged scientifically suitable for publication and will be formally accepted for publication once it meets all outstanding technical requirements.

Kind regards,

Wen-Jun Tu

Academic Editor

PLOS ONE
---

## [Editor Report · Acceptance letter]

2 Jan 2021

PONE-D-20-14697R3 

Is skin disinfection before subcutaneous injection necessary?The reasoning of Certified Nurses in Infection Control in Japan 

Dear Dr. Yoshida:

I'm pleased to inform you that your manuscript has been deemed suitable for publication in PLOS ONE. Congratulations! Your manuscript is now with our production department. 

Kind regards, 

on behalf of

Dr. Wen-Jun Tu 

Academic Editor

PLOS ONE